# POLICY GRADIENT FOR MULTIDIMENSIONAL ACTION SPACES: ACTION SAMPLING AND ENTROPY BONUS

## ABSTRACT

In recent years deep reinforcement learning has been shown to be adept at solving sequential decision processes with high-dimensional state spaces such as in the Atari games. Many reinforcement learning problems, however, involve high-dimensional discrete action spaces as well as high-dimensional state spaces. In this paper, we develop a novel policy gradient methodology for the case of large multidimensional discrete action spaces. We propose two approaches for creating parameterized policies: LSTM parameterization and a Modified MDP (MMDP) giving rise to Feed-Forward Network (FFN) parameterization. Both of these approaches provide expressive models to which backpropagation can be applied for training. We then consider entropy bonus, which is typically added to the reward function to enhance exploration. In the case of high-dimensional action spaces, calculating the entropy and the gradient of the entropy requires enumerating all the actions in the action space and running forward and backpropagation for each action, which may be computationally infeasible. We develop several novel unbiased estimators for the entropy bonus and its gradient. Finally, we test our algorithms on two environments: a multi-hunter multi-rabbit grid game and a multi-agent multi-arm bandit problem.

## 1 INTRODUCTION

In recent years deep reinforcement learning has been shown to be adept at solving sequential decision processes with high-dimensional state spaces such as in the Go game (Silver et al. (2016)) and Atari games (Mnih et al. (2013), Mnih et al. (2015), Mujika (2016), O'Donoghue et al. (2016), Parisotto et al. (2015), Wang et al. (2016), Czarnecki et al. (2017)). In all of these success stories, the size of the action space was relatively small. Many reinforcement learning problems, however, involve high-dimensional action spaces as well as high-dimensional state spaces. Examples include StarCraft (Vinyals et al. (2017), Lin et al. (2017)), where there are many agents each of which can take a finite number of actions; and coordinating self-driving cars at an intersection, where each car can take a finite set of actions (Sukhbaatar et al. (2016)).

In this paper, we develop a novel policy gradient methodology for the case of large multidimensional action spaces. There are two major challenges in developing such a methodology:

- For large multidimensional action spaces, how can we design expressive and differentiable parameterized policies which can be efficiently sampled?

- In policy gradient, in order to encourage sufficient exploration, an entropy bonus term is typically added to the objective function. However, in the case of high-dimensional action spaces, calculating the entropy and its gradient requires enumerating all the actions in the action space and running forward and backpropagation for each action, which may be computationally infeasible. How can we efficiently approximate the entropy and its gradient while maintaining desirable exploration?

In this paper, we first propose two approaches for parameterizing the policy: a LSTM model and a Modified MDP (MMDP) giving rise to Feed-Forward Network (FFN) model. For both of these parameterizations, actions can be efficiently sampled from the policy distribution, and backpropagation can be employed for training. We then develop several novel unbiased estimators for the entropy

bonus and its gradient. These estimators can be combined with stochastic gradient descent giving a new a class of policy gradient algorithms with desirable exploration. Finally, we test our algorithms on two environments: a multi-agent multi-arm bandit problem and a multi-agent hunter-rabbit grid game.

## 2 POLICY GRADIENT FOR MULTIDIMENSIONAL ACTION SPACES

Consider an MDP with a $d$-dimensional action space $\mathcal{A} = \mathcal{A}_1 \times \mathcal{A}_2 \times \cdots \times \mathcal{A}_d$. Denote $\mathbf{a} = (a_1, \ldots, a_d)$ for an action in $\mathcal{A}$. A policy $\pi(\cdot|s)$ specifies for each state $s$ a distribution over the action space $\mathcal{A}$. In the standard RL setting, an agent interacts with an environment over a number of discrete timesteps (Sutton & Barto (1998), Silver (2015)). At timestep $t$, the agent is in state $s_t$ and samples an action $\mathbf{a}_t$ from the policy distribution $\pi(\cdot|s_t)$. The agent then receives a scalar reward $r_t$ and the environment enters the next state $s_{t+1}$. The agent then samples $\mathbf{a}_{t+1}$ from $\pi(\cdot|s_{t+1})$ and so on. The process continues until the end of the episode, denoted by $T$. The return $R_t = \sum_{k=0}^{T-t} \gamma^k r_{t+k}$ is the discounted accumulated return from time step $t$ until the end of the episode.

In the policy gradient formulation, we consider a set of parameterized policies $\pi_\theta(\cdot|s)$, $\theta \in \Theta$, and attempt to find a good $\theta$ within a parameter set $\Theta$. Typically the policy $\pi_\theta(\cdot|s)$ is generated by a neural network with $\theta$ denoting the weights and biases in the network. The parameters $\theta$ are updated by performing stochastic gradient ascent on the expected reward. One example of such an algorithm is REINFORCE, proposed by Williams & Peng (1991), where in a given episode at timestep $t$ the parameters $\theta$ are updated as follows:

$$\Delta\theta = \alpha \sum_{t=0}^{T} \nabla_\theta \log \pi_\theta(\mathbf{a}_t|s_t)(R_t - b_t(s_t))$$

where $b_t(s_t)$ is a baseline. It is well known that the policy gradient algorithm often converges to a local optimum. To discourage convergence to a highly suboptimal policy, the policy entropy is typically added to the update rule:

$$\Delta\theta = \alpha \sum_{t=0}^{T} [\nabla_\theta \log \pi_\theta(\mathbf{a}_t|s_t)(R_t - b_t(s_t)) + \beta \nabla_\theta H_\theta(s_t)] \tag{1}$$

where

$$H_\theta(s_t) := - \sum_{\mathbf{a} \in \mathcal{A}} \pi_\theta(\mathbf{a}|s_t) \log \pi_\theta(\mathbf{a}|s_t) \tag{2}$$

This approach is often referred to as adding entropy bonus or entropy regularization (Williams & Peng (1991)) and is widely used in different applications of neural networks, such as optimal control in Atari games (Mnih et al. (2016)), multi-agent games (Lowe et al. (2017)) and optimizer search for supervised machine learning with RL (Bello et al. (2017)). $\beta$ is referred to as the entropy weight.

In applying policy gradient to MDP with large multidimensional action spaces, there are two challenges. First, how do we design an expressive and differentiable parameterized policy which can be efficiently sampled? Second, for the case of large multidimensional action spaces, calculating the entropy and its gradient requires enumerating all the actions in the action space, which may be infeasible. How do we then enhance exploration in a principled way?

## 3 POLICY PARAMETERIZATION FOR EFFICIENT SAMPLING

To abbreviate the notation, we write $p_\theta(\mathbf{a})$ for $\pi_\theta(\mathbf{a}|s_t)$, with the conditioning on $s_t$ being implicit. We consider schemes whereby the sample components $a_i$, $i = 1, \ldots, d$, are sequentially generated. In particular, after obtaining $a_1, a_2, \ldots, a_{i-1}$, we will generate $a_i \in \mathcal{A}_i$ from some parameterized distribution $p_\theta(\cdot|a_1, a_2, \ldots, a_{i-1})$ defined over the one-dimensional set $\mathcal{A}_i$. After generating the distribution $p_\theta(\cdot|a_1, a_2, \ldots, a_{i-1})$, $i = 1, \ldots, d$ and the action components $a_1, \ldots, a_d$ sequentially, we can then define $p_\theta(\mathbf{a})$ as $p_\theta(\mathbf{a}) = \prod_{i=1}^{d} p_\theta(a_i|a_1, a_2, \ldots, a_{i-1})$. We now propose two methods for creating the parameterized distributions $p_\theta(a|a_1, a_2, \ldots, a_{i-1})$, $a \in \mathcal{A}_i$. To our knowledge, these models are novel and have not been studied in multidimensional action space literature. We assume that the size of the one-dimensional action sets are equal, that is, $|\mathcal{A}_1| = |\mathcal{A}_2| = \ldots = |\mathcal{A}_d| = K$. To handle action sets of different sizes, we include inconsequential actions if needed.

### 3.1 USING RNNS TO GENERATE THE PARAMETERIZED POLICY

The policy $p_\theta(\mathbf{a})$ can be learned with a recurrent neural network (RNN). Long Short-Term Memory (LSTM), a special flavor of RNN, has recently been used with great success to represent conditional probabilities in language translation tasks (Sutskever et al. (2014)). Here, as shown in Figure 1(a), we use an LSTM to generate a parameterized multidimensional distribution $p_\theta(\cdot)$ and to sample $\mathbf{a} = (a_1, \ldots, a_d)$ from that distribution. Specifically, $p_\theta(a|a_1, a_2, \ldots, a_{i-1})$, $a \in \mathcal{A}_i$ is given by the output of the LSTM. To generate $a_i$, we run a forward pass through the LSTM with the input being $a_{i-1}$ and the current state $s_t$ (and implicitly on $a_1, \ldots, a_{i-1}$ which influences $h_{i-1}$). This produces a hidden state $h_i$, which is then passed through a linear layer, producing a $K$ dimensional vector. The softmax of this vector is taken to produce the one-dimensional conditional distribution $p_\theta(a|a_1, a_2, ..., a_{i-1})$, $a \in \mathcal{A}_i$. Finally, $a_i$ is sampled from this one-dimensional distribution, and is then fed into the next stage of the LSTM to produce $a_{i+1}$.

After generating the action $\mathbf{a} = (a_1, \ldots, a_d)$, and the conditional probabilities $p_\theta(\cdot|a_1, a_2, \ldots, a_{i-1})$, $i = 1, \ldots, d$, we can evaluate $p_\theta(\mathbf{a})$ as the product of the conditional probabilities. During training, we can also use backpropagation to efficiently calculate the first term on the RHS of the update rule in (1).

### 3.2 USING MMDP TO GENERATE PARAMETERIZED POLICY

As an alternative to using a LSTM to create parameterized multidimensional policies, we can modify the underlying MDP to create an equivalent MDP for which the action space is one dimensional at each time step. We refer to this MDP as the Modified MDP (MMDP). In the original MDP, we have state space $\mathcal{S}$ and action space $\mathcal{A} = \mathcal{A}_1 \times \mathcal{A}_2 \times \cdots \times \mathcal{A}_d$ where $\mathcal{A}_i = \{1, 2, \ldots, K\}$. In MMDP, the state is modified to encapsulate the original state and all the action dimensions selected for state $s$ so far, i.e., $(s, a_1, a_2, \ldots, a_i, 0, \ldots, 0)$ with $a_1, \ldots, a_i$ being selected values for action dimensions 1 to $i$, and 0 being the placeholder for $d - i - 1$ dimensions. The new action space is $\widetilde{\mathcal{A}} = \{0, 1, \ldots, K\}$ and the new state space is $\mathcal{S} \times \{0, 1, \ldots, K\}^{d-1}$. The state transition probabilities for the MMDP are given by

$$\widetilde{P}((s, a_1, 0, \ldots, 0)|(s, 0, \ldots, 0), a_1) = 1$$
$$\widetilde{P}((s, a_1, a_2, 0, \ldots, 0)|(s, a_1, 0, \ldots, 0), a_2) = 1$$
$$\vdots$$
$$\widetilde{P}((s, a_1, \ldots, a_{d-1})|(s, a_1, \ldots, a_{d-2}, 0), a_{d-1}) = 1$$
$$\widetilde{P}((s', 0, \ldots, 0)|(s, a_1, \ldots, a_{d-1}), a_d) = P(s'|s, a_1, \ldots, a_d)$$

where $P(s'|s, a_1, \ldots, a_d)$ is the transition probabiliy of the original MDP. The reward is only generated after all $d$ component actions are taken. It is easily seen that the MMDP is equivalent to the original MDP.

Since the MMDP has an one-dimensional action space, we can use a feed-forward network (FFN) to generate each action component as shown in (Figure 1(b)). Note that the FFN input layer size is always $|\mathcal{S}| + K - 1$ and the output layer size is $K$.

## 4 ENTROPY BONUS APPROXIMATION FOR LARGE ACTION SPACE

As shown in (1), an entropy bonus is typically included to enhance exploration. However, for large multidimensional action spaces, calculating the entropy and the gradient of the entropy requires enumerating all the actions in the action space and running forward and backpropagation for each action. In this section we develop computationally efficient unbiased estimates for the entropy and its gradient.

Let $\mathbf{A} = (A_1, \ldots, A_d)$ denote a random variable with distribution $p_\theta(\cdot)$. Let $H_\theta$ denote the exact entropy of the distribution $p_\theta(\mathbf{a})$:

$$H_\theta = -\sum_{\mathbf{a}} p_\theta(\mathbf{a}) \log p_\theta(\mathbf{a}) = -\mathbf{E}_{\mathbf{A} \sim p_\theta}[\log p_\theta(\mathbf{A})] = -\sum_{i=1}^{d} \mathbf{E}_{\mathbf{A} \sim p_\theta}[\log p_\theta(A_i|\mathbf{A}_{i-1})]$$

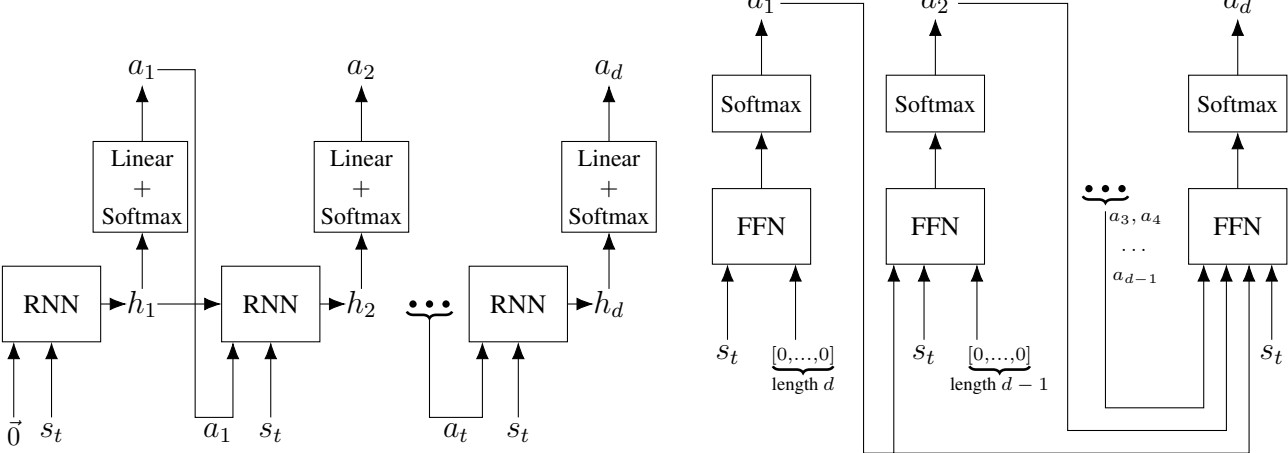

(a) The RNN architecture. To generate $a_i$, we input $s_t$ and $a_{i-1}$ into the RNN and then pass the resulting hidden state $h_i$ through a linear layer and a softmax to generate a distribution, from which we sample $a_i$.

(b) The MMDP architecture. To generate $a_i$, we input $s_t$ and $a_1, a_2, \ldots, a_{i-1}$ into a FFN. The output is passed through a softmax layer, providing a distribution from which we sample $a_i$. Since the input size of the FFN is fixed, when generating $a_i$, constants $0$ serve as placeholders for $a_{i+1}, \ldots, a_{d-1}$ in the input to the FFN.

Figure 1: The RNN and MMDP architectures for generating parameterized policies.

### 4.1 CRUDE UNBIASED ESTIMATOR

During training within an episode, for each state $s_t$, the policy (using, for example, LSTM or MMDP) generates an action $\mathbf{a} = (a_1, a_2, \ldots, a_d)$. A crude approximation of the entropy bonus is:

$$H_\theta^{\text{crude}}(\mathbf{a}) = -\log p_\theta(\mathbf{a}) = -\sum_{i=1}^{d} \log p_\theta(a_i | \mathbf{a}_{i-1})$$

This approximation is an unbiased estimate of $H_\theta$ but its variance is likely to be large. To reduce the variance, we can generate $M$ action samples $\mathbf{a}^{(1)}, \mathbf{a}^{(2)}, \ldots, \mathbf{a}^{(M)}$ when in $s_t$ and average the log action probabilities over the samples. However, generating a large number of samples is costly, especially when each sample is generated from a neural network, since each sample requires one additional forward pass.

### 4.2 SMOOTHED ESTIMATOR

In this section, we develop an alternative unbiased estimator for entropy which only requires the one episodic sample. In the course of an episode, an action $\mathbf{a} = (a_1, a_2, \ldots, a_d)$ is generated for each $s_t$. The alternative estimator accounts for the entropy along each dimension of the action space.

$$\widetilde{H}_\theta(\mathbf{a}) := -\sum_{i=1}^{d} \sum_{a \in \mathcal{A}_i} p_\theta(a | \mathbf{a}_{i-1}) \log p_\theta(a | \mathbf{a}_{i-1}) = \sum_{i=1}^{d} H_\theta^{(i)}(\mathbf{a}_{i-1})$$

where

$$H_\theta^{(i)}(\mathbf{a}_{i-1}) := -\sum_{a \in \mathcal{A}_i} p_\theta(a | \mathbf{a}_{i-1}) \log p_\theta(a | \mathbf{a}_{i-1})$$

which is the entropy of $A_i$ conditioned on $\mathbf{a}_{i-1}$. This approximation of entropy bonus is computationally efficient since for each dimension $i$, we need to obtain $p_\theta(\cdot | \mathbf{a}_{i-1})$, its log and gradient anyway during training. We refer to this approximation as the smoothed entropy.

The smoothed entropy $\widetilde{H}_\theta(\mathbf{A})$ has several appealing properties. The proofs of Theorem 1 and Theorem 3 are straightforward and omitted.

**Theorem 1.** $\widetilde{H}_\theta(\mathbf{A})$ *is an unbiased estimator of the exact entropy* $H_\theta$.

**Theorem 2.** *If $p_\theta(\mathbf{a})$ has a multivariable normal distribution with mean and variance depending on $\theta$, then:*

$$\widetilde{H}_\theta(\mathbf{a}) = H_\theta \quad \forall \mathbf{a} \in \mathcal{A}$$

*Thus, the smoothed entropy equals the exact entropy for a multi-variate normal parameterization of the policy (Proof in Appendix B).*

**Theorem 3.** *(i) If there is a sequence of weights $\theta_1, \theta_2, \ldots$ such that $p_{\theta_n}(\cdot)$ converges to the uniform distribution over $\mathcal{A}$, then*

$$\sup_\theta \widetilde{H}_\theta(\mathbf{a}) = d \log K$$

*(ii) If there is a sequence of weights $\theta_1, \theta_2, \ldots$ such that $p_{\theta_n}(\mathbf{a}^*) \to 1$ for some $\mathbf{a}^*$, then*

$$\inf_\theta \widetilde{H}_\theta(\mathbf{a}) = 0$$

*Thus, the smoothed entropy $\widetilde{H}_\theta(\mathbf{a})$ mimics the exact entropy in that it has the same supremum and infinum values as the exact entropy.*

The above theorems indicate that $\widetilde{H}_\theta(\mathbf{a})$ may serve as a good proxy for $H_\theta$: it is an unbiased estimator for $H_\theta$, it has the same minimum and maximum values when varying $\theta$; and in the special case when $p_\theta(\mathbf{a})$ has a multivariate normal distribution, it is actually equal to $H_\theta$ for all $\mathbf{a} \in \mathcal{A}$. Our numerical experiments have shown that the smoothed estimator $\widetilde{H}_\theta(\mathbf{a})$ typically has lower variance than the crude estimator $H_\theta^{\text{crude}}(\mathbf{a})$. However, it is not generally true that the smoothed estimator always has lower variance as counterexamples can be found.

## 4.3 GRADIENT OF ENTROPY

So far we have been looking at estimates of entropy. But the policy gradient algorithm (1) uses the gradient of the entropy rather than just simply the entropy. As it turns out, the gradient of estimators $H_\theta^{\text{crude}}(\mathbf{a})$ and $\widetilde{H}_\theta(\mathbf{a})$ are not unbiased estimates of the gradient of the entropy. In this subsection, we provide unbiased estimators for the gradient of the entropy. For simplicity, in this section, we assume an one-step decision setting, such as in a multi-armed bandit problem. A straightforward calculation shows:

$$\nabla_\theta H_\theta = \mathrm{E}_{\mathbf{A} \sim p_\theta}[-\log p_\theta(\mathbf{A}) \nabla_\theta \log p_\theta(\mathbf{A})] \tag{3}$$

Suppose $\mathbf{a}$ is one sample from $p_\theta(\cdot)$. A crude unbiased estimator for the gradient of the entropy therefore is: $-\log p_\theta(\mathbf{a}) \nabla_\theta \log p_\theta(\mathbf{a}) = \log p_\theta(\mathbf{a}) \nabla_\theta H_\theta^{\text{crude}}(\mathbf{a})$. Note that this estimator is equal to the gradient of the crude estimator multiplied by a correction factor.

Analogous to the smoothed estimator for entropy, we can also derive a smoothed estimator for the gradient of the entropy.

**Theorem 4.** *If $\mathbf{a}$ is a sample from $p_\theta(\cdot)$, then*

$$\nabla_\theta \widetilde{H}_\theta(\mathbf{a}) + \sum_{i=1}^{d} H_\theta^{(i)}(\mathbf{a}_{i-1}) \nabla_\theta \sum_{j=1}^{i-1} \log p_\theta(a_j | \mathbf{a}_{j-1})$$

*is an unbiased estimator for the gradient of the entropy (Proof in Appendix C).*

Note that this estimate for the gradient of the entropy is equal to the gradient of the smoothed estimate $\widetilde{H}_\theta(\mathbf{a})$ plus a correction term. We refer to this estimate of the entropy gradient as the unbiased gradient estimate.

## 5 EXPERIMENTAL RESULTS

We designed experiments to compare the LSTM and MMDP models, and to also compare how the different entropy approximations perform for both. For each entropy approximation, the entropy weight as described in (1) was tuned to give the highest episode reward. For MMDP, the number of hidden layers was also tuned from 1 to 7. The rest of the hyperparameters are listed in Appendix A.

### 5.1 Hunters and Rabbits

In this environment, there is a $n \times n$ grid. At the beginning of each episode $d$ hunters and $d$ rabbits are randomly placed in the grid. The rabbits remain fixed in the episode, and each hunter can move to a neighboring square (including diagonal neighbors) or stay at the current square. So each hunter has nine possible actions, and altogether there are $|\mathcal{A}| = 9^d$ actions at each time step. When a hunter enters a square with a rabbit, the hunter captures the rabbit and remains there until the end of the game. In each episode, the goal is for the hunters to capture the rabbits as quickly as possible. Each episode is allowed to run for at most 10,000 time steps.

To provide a dense reward signal, we formalize the goal with the following modification: capturing a rabbit gives a reward of 1, which is discounted by the number of time steps taken since the beginning of the episode. The discount factor is 0.8. The goal is to maximize the episode's total discounted reward. After a hunter captures a rabbit, they both become inactive. The representation of an active hunter or rabbit is (1, y position, x position). The representation of an inactive hunter or rabbit is (0, -1, -1).

*Comparison of different entropy estimates for LSTM and MMDP*

Table 1 shows the performance of the LSTM and MMDP models with different entropy estimates. (smoothed mode entropy is explained in Appendix D). The evaluation was performed in a square grid of 5 by 5 with 5 hunters and 5 rabbits. Training was run for 1 million episodes for each of the seeds. All evaluations are averaged over 1,000 episodes per seed for a total of 5,000 episodes.

First, we observe that the LSTM model always does better than the MMDP model, particularly for the episode length. Second, we note that policies obtained with the entropy approximations all perform better than policies obtained without entropy or with crude entropy. For the LSTM model, the best performing approximation is smoothed entropy, reducing the mean episode length by $45\%$ and increasing the mean episode reward by $10\%$ compared to without entropy. We also note that there is not a significant difference in performance between the smoothed entropy estimate, smoothed mode estimate, and the unbiased gradient estimate.

Table 1: Performance of LSTM and MMDP across different entropy approximations.

|  | Without Entropy | Crude Entropy | Smoothed Entropy | Smoothed Mode Entropy | Unbiased Gradient Estimate |
|---|---|---|---|---|---|
| LSTM Mean Episode Length | $10.1 \pm 1.9$ | $19 \pm 8.7$ | $5.6 \pm 0.2$ | $6.0 \pm 0.2$ | $6.0 \pm 0.1$ |
| MMDP Mean Episode Length | $21.5 \pm 3.7$ | $37.3 \pm 29.6$ | $10.6 \pm 0.7$ | $10.6 \pm 0.7$ | $9.8 \pm 0.6$ |
| LSTM Mean Episode Reward | $3.0 \pm 0.06$ | $3.0 \pm 0.03$ | $3.3 \pm 0.04$ | $3.2 \pm 0.04$ | $3.2 \pm 0.02$ |
| MMDP Mean Episode Reward | $2.8 \pm 0.03$ | $2.7 \pm 0.03$ | $2.9 \pm 0.03$ | $2.8 \pm 0.04$ | $2.9 \pm 0.02$ |

As shown in Table 2, smoothed entropy is also more robust to the initial seed than without entropy. For example, for the LSTM model, in the case of without entropy, seed 0 leads to significantly worse results than the seeds 1-4. This does not happen to smoothed entropy.

*Entropy approximations versus exact entropy*

We now consider how policies trained with entropy approximations compare with polices trained with exact entropy. In order to calculate exact entropy in an acceptable amount of time, we reduced the number of hunters and rabbits to 4 hunters and 4 rabbits. Training was run for 50,000 episodes. Table 3 shows the performance differences between policies trained with entropy approximations and exact entropy. We see that the best entropy approximations perform only slightly worse than exact entropy for both LSTM and MMDP. Once again we see that the LSTM model performs better than the MMDP model.

Table 2: LSTM and MMDP results across seeds.

| | Without Entropy | | | | | Crude Entropy | | | | | Smoothed Entropy | | | | |
|---|---|---|---|---|---|---|---|---|---|---|---|---|---|---|---|
| Seed | 0 | 1 | 2 | 3 | 4 | 0 | 1 | 2 | 3 | 4 | 0 | 1 | 2 | 3 | 4 |
| LSTM Mean Episode Length | 14 | 9 | 11 | 9 | 8 | 40 | 12 | 17 | 11 | 14 | 5 | 6 | 6 | 5 | 6 |
| MMDP Mean Episode Length | 15 | 19 | 27 | 27 | 20 | 17 | 30 | 14 | 109 | 18 | 10 | 10 | 11 | 11 | 12 |

Table 3: LSTM and MMDP results for entropy approximation versus exact entropy.

| | LSTM Smoothed Entropy | LSTM Exact Entropy | MMDP Unbiased Gradient Estimate | MMDP Exact Entropy |
|---|---|---|---|---|
| Mean Episode Length | $9.0 \pm 0.3$ | $8.9 \pm 0.2$ | $11.5 \pm 0.3$ | $10.7 \pm 0.4$ |
| Mean Episode Reward | $2.14 \pm 0.02$ | $2.19 \pm 0.02$ | $2.01 \pm 0.01$ | $2.1 \pm 0.01$ |

## 5.2 MULTI-AGENT MULTI-ARM BANDITS

We examine a multi-agent version of the standard multi-armed bandit problem, where there are $d$ agents each pulling one of $K$ arms, with $d \leq K$. The $k^{th}$ arm generates a reward $r_k$. The total reward in a round is generated as follows. In each round, each agent chooses an arm. All of the chosen arms are then pulled, with each pulled arm generating a reward. Note that the total number of arms chosen, $c$, may be less than $d$ since some agents may choose the same arm. The total reward is the sum of rewards from the $c$ chosen arms. The optimal policy is for the $d$ agents to collectively pull the $d$ arms with the highest rewards. Additionally, among all the optimal assignments of $d$ agents to the $d$ arms that yield the highest reward, we add a bonus reward with probability $p^*$ if one particular agent-to-arms assignment is chosen.

We performed experiments with 4 agents and 10 arms, with the $k^{th}$ arm providing a reward of $k$. The exceptional assignment gets a bonus of 200 with probability 0.01, and no bonus with probability 0.99. Thus the maximum expected reward is 36. Training was run for 100,000 rounds for each of the seeds. Table 4 shows average results for the last 500 of the 100,000 rounds.

Table 4: Performance of LSTM policy parameterization.

| | Without Entropy | Crude Entropy | Smoothed Entropy | Unbiased Gradient Estimate |
|---|---|---|---|---|
| Mean Reward | $34.9 \pm 0.8$ | $35.5 \pm 1.1$ | $35.9 \pm 0.8$ | $35.9 \pm 0.3$ |
| Percentage Optimal Arms Pulled | $39.8 \pm 35.9$ | $59.4 \pm 35.7$ | $95.0 \pm 1.9$ | $95.7 \pm 2.7$ |

The results for the multi-agent bandit problem are consistent with those for the hunter-rabbit problem. Policies obtained with the entropy approximations all perform better than policies obtained without entropy or with crude entropy, particularly for the percentage of optimal arms pulled. We again note that using the unbiased gradient estimate does not perform significantly better than using the smoothed entropy estimate.

## 6 RELATED WORK

There has been limited attention in the RL literature with regards to large discrete action spaces. Pazis & Parr (2011) proposes generalized value functions in the form of H-value functions, and also propose approximate linear programming as a solution technique. Their methodology is not suited for deep RL, and approximate linear programming may lead to highly sub-optimal solutions.

Dulac-Arnold et al. (2015) embeds discrete actions in a continuous space, picks actions in the continuous space and map these actions back into the discrete space. However, their algorithm introduces a new hyper-parameter that requires tuning for every new task. Our approach involves no new hyper-parameter other than those normally used in deep learning.

In Sukhbaatar et al. (2016), each action dimension is treated as an agent and backpropagation is used to learn coordination between the agents. The approach is particularly adept for problems where agents leave and enter the system. However, the approach requires homogenous agents, and has not been shown to solve large-scale problems. Furthermore, the decentralized approach will potentially lead to highly suboptimal polices even though communication is optimized among the agents.

To our knowledge, we are the first to propose using LSTMs and a modified MDP to create policies for RL problems with large multidimensional action spaces. Although this leads to algorithms that are straightforward, the approaches are natural and well-suited to multidimensional action spaces.

We also propose novel estimators for the entropy regularization term that is often used in policy gradient. To the best of our knowledge, no prior work has dealt with approximating the policy entropy for MDP with large multidimensional discrete action space. On the other hand, there has been many attempts to devise methods to encourage beneficial exploration for policy gradient. Nachum et al. (2016) modifies the entropy term by adding weights to the log action probabilities, leading to a new optimization objective termed under-appreciated reward exploration.

While entropy regularization has been mostly used in algorithms that explicitly parameterize the policies, Schulman et al. (2017) applies entropy regularization to Q-learning methods. They make an important observation about the equivalence between policy gradient and entropy regularized Q-learning, which they term soft Q-learning.

## 7 CONCLUSION

In this paper, we developed a novel policy gradient methodology for the case of large multidimensional discrete action spaces. We proposed two approaches for creating parameterized policies: LSTM parameterization and a Modified MDP (MMDP) giving rise to Feed-Forward Network (FFN) parameterization. Both of these approaches provide expressive models to which backpropagation can be applied for training. We then developed several novel unbiased estimators for entropy bonus and its gradient. We did experimental work for two environments with large multidimensional action space. For these environments, we found that both the LSTM and MMDP approach could successfully solve large multidimensional action space problems, with the LSTM approach generally performing better. We also found that the smoothed estimates of the entropy and the unbiased gradient estimate of the entropy gradient can help reduce computational cost while not sacrificing significant loss in performance.

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

## APPENDIX

## A    HYPERPARAMETERS

*Hyperparameters for hunter-rabbit game*

The LSTM policy has 128 hidden nodes. For the MMDP policy, the number of hidden layers for smoothed entropy, smoothed mode entropy, unbiased gradient estimate, crude entropy and without entropy are 5, 3, 3, 4 3 and 3 respectively. Each MMDP layer has 128 nodes. We parameterize the baseline in (1) with a feed forward neural network with one hidden layer of size 64. This network was trained using first visit Monte Carlo return to minimize the L1 loss between actual and predicted values of states visited during the epidode.

Both the policies and baseline are optimized after each episode with RMSprop (Tieleman & Hinton (2012)). The RHS of (1) is clipped to $[-1, 1]$ before updating the policy parameters. The learning rates for the baseline, LSTM and MMDP are $10^{-3}$, $10^{-3}$, $10^{-4}$ respectively.

To obtain the results in Table 1, the entropy weights for LSTM smoothed entropy, LSTM smoothed mode entropy, LSTM unbiased gradient estimate, LSTM crude entropy, MMDP smoothed entropy, MMDP smoothed mode entropy, MMDP unbiased gradient estimate and MMDP crude entropy are 0.02, 0.021, 0.031, 0.04, 0.02, 0.03, 0.03 and 0.01 respectively.

To obtain the results in Table 3, the entropy weights for LSTM smoothed entropy, LSTM exact entropy, MMDP unbiased gradient estimate and MMDP exact entropy are 0.03, 0.01, 0.03 and 0.01 respectively. The MMDP networks have three layers with 128 nodes in each layer. Experimental results are averaged over five seeds.

*Hyperparamters for Multi-Agent Multi-Arm Bandits*

The experiments were run with 4 agents and 10 arms. For the 10 arms, their rewards are $i$ for $i = 1, \ldots, 10$. The LSTM policy has 32 hidden nodes. The baseline in (1) is a truncated average of the reward of the last 100 rounds. The entropy weight for crude entropy, smoothed entropy and unbiased gradient estimate are 0.005, 0.001 and 0.003 respectively. The learning rates for without entropy, crude entropy, smoothed entropy and unbiased gradient estimate are 0.006, 0.008, 0.002 and 0.005 respectively. Experimental results are averaged over ten seeds.

## B    PROOF OF THEOREM 2

We first note that for $\begin{bmatrix} \mathbf{X}_1 \\ \mathbf{X}_2 \end{bmatrix} \sim N\left( \begin{bmatrix} \boldsymbol{\mu}_1 \\ \boldsymbol{\mu}_2 \end{bmatrix}, \begin{bmatrix} \boldsymbol{\Sigma}_{11} & \boldsymbol{\Sigma}_{12} \\ \boldsymbol{\Sigma}_{21} & \boldsymbol{\Sigma}_{22} \end{bmatrix} \right)$ where $\mathbf{X}_1$ and $\mathbf{X}_2$ are random vectors, we have $\mathbf{X}_2 \mid \mathbf{X}_1 = \mathbf{x}_1 \sim N(\bar{\boldsymbol{\mu}}, \bar{\boldsymbol{\Sigma}})$ where

$$\bar{\boldsymbol{\mu}} = \boldsymbol{\mu}_2 + \boldsymbol{\Sigma}_{21}\boldsymbol{\Sigma}_{11}^{-1}(\mathbf{x}_1 - \boldsymbol{\mu}_1)$$
$$\bar{\boldsymbol{\Sigma}} = \boldsymbol{\Sigma}_{22} - \boldsymbol{\Sigma}_{21}\boldsymbol{\Sigma}_{11}\boldsymbol{\Sigma}_{12}$$

Observe that the covariance matrix of the conditional distribution does not depend on the value of $x_1$ (Johnson & Wichern (1988)).

Also note that for $\mathbf{X} \sim N(\boldsymbol{\mu}, \boldsymbol{\Sigma})$, the entropy of $\mathbf{X}$ takes the form

$$H(\mathbf{X}) = \frac{k}{2}(\log 2\pi + 1) + \frac{1}{2}|\boldsymbol{\Sigma}|$$

where $k$ is the dimension of $\mathbf{X}$ and $|\cdot|$ denotes the determinant. Therefore, the entropy of a multivariate normal random variable depends only on the variance and not on the mean.

Because $\mathbf{A}$ is multivariate normal, the distribution of $A_i$ given $A_1 = a_1, \ldots, A_{i-1} = a_{i-1}$ has a normal distribution with a variance $\sigma_i^2$ that does not depend on $a_1, \ldots, a_{i-1}$. Therefore

$$H_\theta(A_i|a_1, \ldots, a_{i-1}) = \frac{1}{2}(\log 2\pi + 1 + \sigma_i^2)$$

does not depend on $a_1, \ldots, a_{i-1}$ and hence $\widetilde{H}_\theta(\mathbf{a})$ does not depend on $\mathbf{a}$. Combining this with the fact that $\widetilde{H}_\theta(\mathbf{a})$ is an unbiased estimator for $H_\theta$ gives $\widetilde{H}_\theta(\mathbf{a}) = H_\theta$ for all $\mathbf{a} \in \mathcal{A}$.

## C    PROOF OF THEOREM 4

From (3), we have:

$$\nabla_\theta H_\theta = -\sum_{i=1}^{d} \sum_{j=1}^{d} \mathrm{E}_{\mathbf{A} \sim p_\theta}[\log p_\theta(A_i|\mathbf{A}_{i-1}) \nabla_\theta \log p_\theta(A_j|\mathbf{A}_{j-1})] \qquad (4)$$

We will now use conditional expectation to calculate the terms in the double sum.

For $i < j$:

$$\mathrm{E}_{\mathbf{A} \sim p_\theta}[\log p_\theta(A_i|\mathbf{A}_{i-1}) \nabla_\theta \log p_\theta(A_j|\mathbf{A}_{j-1})|\mathbf{A}_{j-1}]$$
$$= \log p_\theta(A_i|\mathbf{A}_{i-1}) \mathrm{E}_{\mathbf{A} \sim p_\theta}[\nabla_\theta \log p_\theta(A_j|\mathbf{A}_{j-1})|\mathbf{A}_{j-1}] = 0$$

For $i > j$:

$$\mathrm{E}_{\mathbf{A} \sim p_\theta}[\log p_\theta(A_i|\mathbf{A}_{i-1}) \nabla_\theta \log p_\theta(A_j|\mathbf{A}_{j-1})|\mathbf{A}_{i-1}]$$
$$= \nabla_\theta \log p_\theta(A_j|\mathbf{A}_{j-1}) \mathrm{E}_{\mathbf{A} \sim p_\theta}[\log p_\theta(A_i|\mathbf{A}_{i-1})]$$
$$= -\nabla_\theta \log p_\theta(A_j|\mathbf{A}_{j-1}) H_\theta^{(i)}(\mathbf{A}_{i-1})$$

For $i = j$:

$$\mathrm{E}_{\mathbf{A} \sim p_\theta}[\log p_\theta(A_i|\mathbf{A}_{i-1}) \nabla_\theta \log p_\theta(A_i|\mathbf{A}_{i-1})|\mathbf{A}_{i-1}] = -\nabla_\theta H_\theta^{(i)}(\mathbf{A}_{i-1})$$

Combining these three conditional expectations with (4), we obtain:

$$\nabla_\theta H_\theta = \mathrm{E}_{\mathbf{A} \sim p_\theta}[\nabla_\theta \widetilde{H}_\theta(\mathbf{A}) + \sum_{i=1}^{d} H_\theta^{(i)}(\mathbf{A}_{i-1}) \nabla_\theta \sum_{j=1}^{i-1} \log p_\theta(A_j|\mathbf{A}_{j-1})]$$

## D    APPROXIMATING ENTROPY USING THE MODE OF THE DISTRIBUTION

Depending on the episodic action $\mathbf{a}$ at a given time step in the episode, the smoothed entropy $\widetilde{H}_\theta(\mathbf{a})$ may give unsatisfactory results. For example, suppose for a particular episodic action $\mathbf{a}$, $\widetilde{H}_\theta(\mathbf{a}) \gg H_\theta$. In this case, the policy gradient may ignore the entropy bonus term, thinking that the policy already has enough entropy when it perhaps does not. We therefore consider alternative approximations which may improve performance at modest additional computational cost.

First consider

$$E^* = -\sum_{i=1}^{d} \sum_{a \in \mathcal{A}_i} p_\theta(a|a_1^*, \ldots, a_{i-1}^*) \log p_\theta(a|a_1^*, \ldots, a_{i-1}^*)$$

where

$$\mathbf{a}^* = (a_1^*, \ldots, a_d^*) = \underset{\mathbf{a} \in \mathcal{A}}{\operatorname{argmax}}\, p_\theta(\mathbf{a})$$

Thus in this case, instead of calculating the entropy over a sample action $\mathbf{a}$, we calculate it over the most likely action $\mathbf{a}^*$. The problem here is that it is not easy to find $\mathbf{a}^*$ when the given conditional probabilities $p_\theta(a|a_1, \ldots, a_{i-1})$ are not in closed form but only available algorithmically as outputs of neural networks.

A more computationally efficient approach would be to choose the action greedily:

$$\hat{a}_1 = \underset{a \in \mathcal{A}_1}{\operatorname{argmax}}\, p_\theta(a)$$

$$\hat{a}_2 = \underset{a \in \mathcal{A}_2}{\operatorname{argmax}}\, p_\theta(a|\hat{a}_1)$$

$$\vdots$$

$$\hat{a}_{d-1} = \underset{a \in \mathcal{A}_{d-1}}{\operatorname{argmax}}\, p_\theta(a|\hat{a}_1, \ldots, \hat{a}_{d-2})$$

This leads to the definition

$$\widehat{H}_\theta = -\sum_{i=1}^{d} \sum_{a \in \mathcal{A}_i} p_\theta(a|\hat{a}_1, \ldots, \hat{a}_{i-1}) \log p_\theta(a|\hat{a}_1, \ldots, \hat{a}_{i-1})$$

The action $\hat{\mathbf{a}}$ is an approximation for the mode of the distribution $p_\theta(\cdot)$. As often done in NLP, we can use beam search to determine an action $\mathbf{a}'$ that is a better approximation, that is, $p_\theta(\mathbf{a}') \geq p_\theta(\hat{\mathbf{a}})$. Indeed, the above $\widehat{H}_\theta$ definition is beam search with beam size equal to 1. We refer to $\widehat{H}_\theta$ as smoothed mode entropy.

$\widehat{H}_\theta$ with an appropriate beam size may be a better approximation for the entropy $H_\theta$ than $\widetilde{H}_\theta(\mathbf{a})$. However, calculating $\widehat{H}_\theta$ and its gradient comes with some computational cost. For example, with a beam size equal to one, we would have to make two passes through the neural network at each time step: one to obtain the episodic sample $\mathbf{a}$ and the other to obtain the greedy action $\hat{\mathbf{a}}$. For beam size $n$ we would need to make $n + 1$ passes.

We note that $\widehat{H}_\theta$ is a biased estimator for $H_\theta$ but with no variance. Thus there is a bias-variance tradeoff between $\widetilde{H}_\theta(\mathbf{a})$ and $\widehat{H}_\theta$. Note that $\widehat{H}_\theta$ also satisfies Theorems 2 and 3 in subsection 4.2.

