# OpenReview forum: "Policy Gradient For Multidimensional Action Spaces: Action Sampling and Entropy Bonus"
_ICLR.cc/2018/Conference — Reject_

### Official Review · AnonReviewer3 · 2017-11-27

**Rating:** 6
**Confidence:** 3

**Review:**

In this paper, the authors suggest introducing dependencies between actions in RL settings with multi-dimensional action spaces by way of two mechanisms (using an RNN and making partial action specification as part of the state); they then introduce entropy pseudo-rewards whose maximization corresponding to joint entropy maximization.

In general, the multidimensional action methods either seem incremental or non novel to me. The combined use of the chain rule and RNNs (LSTM or not) to induce correlations in multi-dimensional outputs is well know (sequence-to-sequence networks, pixelRNN, etc.) and the extension to RL presents no difficulties, if it is not already known. Note very related work in https://arxiv.org/pdf/1607.07086.pdf and https://www.media.mit.edu/projects/improving-rnn-sequence-generation-with-rl/overview/ .

As for the MMDP technique, I believe it is folklore (it can for instance be found as a problem in a problem set - http://stellar.mit.edu/S/course/2/sp04/2.997/courseMaterial/topics/topic2/readings/problemset4/problemset4.pdf). Note that both approaches could be combined; the first idea is essentially a policy method, the second, a value method. The second method could be used to provide stronger, partial action-conditional baselines (or even critics) to the first method.

The entropy derivation are more interesting - and the smoothed entropy technique is as far as I know, novel. The experiments are well done, though on simple toy environments.

Minor:
- In section 3.2, one should in principle tweak the discount factor of the modified MDP to recover behavior identical to the original one with large action space. This should be noted (alternatively, the discount between non-environment transitions should be set to 1).

- From the description at the end of 3.2, and figure 1.b, it seems actions fed to the MMDP feed-forward network are not one-
hot; I thought this was pretty surprising as it would almost certainly affect performance? Note also that the collection of feed-forward network which collectively output the joint vector can be thought of as an RNN with non-learned state transition.

- Since the function optimized can be written as an expectation of reward+pseudo-reward, the proof of theorem 4 can be simplified by using generic score-function optimization arguments (see Stochastic Computation Graphs, Schulman et al).

---

> ### Author Response · Authors · 2018-01-04
> **Thank you for your review!**
>
> Thank you for your helpful pointers to the relevant LSTM and MMDP literature. In light of your review, we have rewritten our paper to focus on the novel entropy estimates and properly acknowledge relevant previous works. We believe this new emphasis has led to a substantially improved paper.
>
> As you requested, in the revised paper, we noted that for Modified MDP, the discount between non-environment transitions should be set to 1 to match the original MDP (which is what we did in our experiments).
>
> As you requested, we also tried representing the actions fed to the Modified MDP feed-forward network as one-hot vectors. We noted in Appendix D that the one-hot vectors did not bring substantial improvement.
>
> Finally, we took a close look at the Schulman et al paper for proving our result for the smoothed gradient entropy estimator. Although with this approach the "proof" would just be a couple lines, to justify using the proof rigorously would require a lengthy explanation on how to fit our model into the model of Schulman et al. We have, however, indicated that the result could be alternatively proven using Theorem 1 of Schulman et al and provided a reference.

---

### Official Review · AnonReviewer2 · 2017-11-27
**Simple autoregressive model for action spaces, but missing some baselines**

**Rating:** 5
**Confidence:** 4

**Review:**

The authors present two autoregressive models for sampling action probabilities from a factorized discrete action space. On a multi-agent gridworld task and a multi-agent multi-armed bandit task, the proposed method seems to benefit from their lower-variance entropy estimator for exploration bonus. A few key citations were missing - notably the LSTM model they propose is a clear instance of an autoregressive density estimator, as in PixelCNN, WaveNet and other recently popular deep architectures. In that context, this work can be viewed as applying deep autoregressive density estimators to policy gradient methods. At least one of those papers ought to be cited. It also seems like a simple, obvious baseline is missing from their experiments - simply independently outputting D independent softmaxes from the policy network. Without that baseline it's not clear that any actual benefit is gained by modeling the joint distribution between actions, especially since the optimal policy for an MDP is provably deterministic anyway. The method could even be made to capture dependencies between different actions by adding a latent probabilistic layer in the middle of the policy network, inducing marginal dependencies between different actions. A direct comparison against one of the related methods in the discussion section would help better contextualize the paper as well. A final point on clarity of presentation - in keeping with the convention in the field, the readability of the tables could be improved by putting the top-performing models in bold, and Table 2 should almost certainly be replaced by a boxplot.

---

> ### Author Response · Authors · 2018-01-04
> **Thank you for your review!**
>
> In our revision, we acknowledge that the LSTM policy parameterization is not entirely new and can, in fact, be seen as an adaption of auto-regressive techniques in supervised sequence modeling to reinforcement learning (Sections 4.3 and 6). We have reorganized our paper to focus on the novel entropy estimates.
>
> As you requested, we added experimental results for the baseline for which the policy is a FFN with multiple heads. We refer to this as Independent Sampling (IS). We ran experiments for IS with and without (estimates of) the entropy bonus for both the rabbit and bandit environments.
>
> You also suggested that we compare our results with one of the approaches in the literature. For this, we choose the paper “Learning Multiagent Communication with Backpropagation”. Please find the results in Section 5. As you requested, we also put the top-performing models in bold and turned Table 2 into a boxplot (Table 2 is now Figure 2).

---

### Official Review · AnonReviewer1 · 2017-11-30
**The paper deals with the problem of large-multi-dimensional action space in RL. It proposes an auto-regressive model to represent the policy, in which the value of action at each dimension will be represented as a function of state and the "previous" dimensions. I think the idea is very interesting and  useful but it is already explored in the context of Deep RL before. So It is not entirely novel contrary to the authors claim.**

**Rating:** 5
**Confidence:** 5

**Review:**

Clarity and quality:

The paper is well written and the ideas are motivated clearly both in writing and with block diagram panels.  Also the fact  that the paper considers different  variants of  the idea  adds to the quality of the paper. May main concern is with the quality of results which is limited to some toy/synthetic problems. Also the comparison with the previous work is missing.The paper would benefit from  a more in depth numerical analysis of this approach both by applying it to more challenging/standard domains such as Mujoco and also by comparing the results with prior approaches such as A3C, DDPG and TRPO.

Originality, novelty and Significance:

The paper claims that the approach is novel in the context of policy gradient and Deep RL. I am not sure this is entirely the case since there is a recent work from Google Brain (https://arxiv.org/pdf/1705.05035.pdf ) which consider almost the identical idea with the same variation in the context of DQN and policy gradient (they call their policy gradient approach  Prob SDQN).  The Brain paper also  makes a much more convincing case with their numerical analysis, applied to more challenging domains such as control suite. The paper under review  should acknowledge this prior work and discuss the similarities and the differences. Also since the main idea and the algorithms are quiet similar to the Brain paper I believe the novelty of this work is at best marginal.

---

> ### Author Response · Authors · 2018-01-04
> **Thank you for your review!**
>
> We have revised our paper to acknowledge the Google Brain paper in the formulation of the MMDP and the LSTM policy parameterizations (Sections 4 and 6). We have also acknowledged other relevant previous work on autoregressive models for policy gradient.
>
> Our paper differs from the Google Brain paper with regards to exploration strategies. Whereas the Brain paper injects noise into the action space to encourage exploration, the focus of our paper is to develop novel unbiased estimates for the entropy bonus and its gradient.
>
> We put great effort into trying to apply our approach to the Mujoco domain. However, we faced technical challenges and thus could not complete it in time. For example, the OpenAI Mujoco interface, which uses Mujoco 1.3.1, is incompatible with our workstations, which are Macs with NVMe disks. For more info on the issue, please have a look at the links below:
>
> https://github.com/openai/mujoco-py/issues/36
> http://www.mujoco.org/forum/index.php?threads/error-could-not-open-disk.3441/
>
> We also had issues compiling Mujoco and its dependencies on our HPC, such as the Mesa 3D Graphics Library. Although we were not able to run experiments in the more complex Mujoco environments, we believe that the simplicity of the environments used in our paper help to highlight critical issues related to entropy bonus.
>
> Thank you for your pointer to A3C, DDPG and TRPO. Our entropy estimators are orthogonal to these approaches and thus they potentially can be combined with them. We may explore the benefits of our entropy estimates for these approaches in future work.

---

### Author Response · Authors · 2018-01-04
**General reply to all reviewers and the list of changes in the revised paper (4th Jan 2018)**

We would like to thank all three reviewers for your detailed reviews and useful insights. Your comments have led to a greatly improved revised paper, without substantially changing the content of the original paper.

One consensus among the reviewers seems to be that although the material on entropy estimates is novel and interesting, the material on autoregressive models (MMDP and LSTM) is less novel since this material was already known in the folklore or presented in recent papers. You also requested that in addition to the autoregressive models, we examine other baseline policies.

Responding to these concerns, we have re-organized the material to place more emphasis on the entropy estimates and less emphasis on the autoregressive policy models. In doing so, we have cited earlier work on using MMDP and LSTM for policy gradient, including the recent Google Brain paper. We also made a major effort to generate results for two baseline policies: (1) a single feed-forward network with multiple heads; and (2) CommNet. For both of these baselines, we examined our various entropy estimates.

Please note that with the exception of new experimental results, there is no new material presented in our revised paper. However, the paper has undergone a major reorganization. The list of changes is below:

- The title of the paper is changed to reflect the new emphasis on the entropy estimates.

- The abstract and introduction are rewritten to focus on the entropy estimates and removed the claim that the autoregressive policies are novel.

- The entropy estimates section is moved to before the policy parameterization section.

- The smoothed mode estimator is moved from the appendix to the entropy estimates section (now subsection 3.3).

- In the policy parameterization section, a new subsection 4.1 is added to explain the new baseline policy (1) above.

- The MMDP subsection is shortened to present the minimal explanation and the details are moved to Appendix D. The MMDP subsection acknowledges prior work by the Google Brain paper.

- The LSTM subsection acknowledges relevant prior work.

- In the experimental results section, we introduce CommNet and added results for baseline policies (1), (2) for the hunter-rabbit environment and added results for baseline policy (1) for multi-arm bandits.

- The hunter-rabbit result analysis is rewritten to place more emphasis on the role of the entropy estimates across different models for the policy.

- In Table 1, best performing models are bolded and horizontal lines are added to improve readability.

- Table 2 is turned into a boxplot and is now Figure 2.

- In Related Work section, we first discussed relevant work with regards to the entropy estimates before the policy parameterizations.

- The Conclusion is shortened so the paper stays within the recommended 8-page length.

- The hyperparameters for the two baseline policies are added to Appendix A.

- At the end of the proof of theorem 4 in Appendix C, we note that the theorem can also be proved by material introduced by Stochastic Computational Graph of Schulman et al and provided a reference.

- Appendix D is added to explain the details of MMDP. We noted that the discount between non-environment transitions should be set to 1 to match the original MDP and that we tried representing the actions fed to the Modified MDP feed-forward network as one-hot vectors.

- Appendix E is added to explain the state representation for baseline policy (2).

- Other minor changes to improve readability.

---

### Decision · Program_Chairs · 2018-01-29
**ICLR 2018 Conference Acceptance Decision**

**Decision:**

Reject

**Comment:**

The paper has some interesting ideas around auto-regressive policies and estimating their entropy for exploration. The use of autoregressive policies in RL is not particularly novel, and the estimate of entropy for such models is straightforward. Finally, the experiments focus on very simple tasks.